# Long-Term Prognosis of Febrile Individuals with Right Precordial Coved-Type ST-Segment Elevation Brugada Pattern: A 10-Year Prospective Follow-Up Study

**DOI:** 10.3390/jcm10214997

**Published:** 2021-10-27

**Authors:** Chin-Feng Tsai, Yao-Tsung Chuang, Jing-Yang Huang, Kwo-Chang Ueng

**Affiliations:** 1Division of Cardiology, Department of Internal Medicine, Chung Shan Medical University Hospital, Taichung 40201, Taiwan; 2Institute of Medicine, School of Medicine, Chung Shan Medical University, Taichung 40201, Taiwan; force118chuang@gmail.com (Y.-T.C.); kcueng@gmail.com (K.-C.U.); 3Department of Medical Research, Chung Shan Medical University Hospital, Taichung 40201, Taiwan; wchinyang@gmail.com

**Keywords:** Brugada syndrome, electrocardiogram, fever, genetic disorder, sudden cardiac death, ventricular arrhythmia

## Abstract

A febrile state may provoke a Brugada electrocardiogram (ECG) pattern and trigger ventricular tachyarrhythmias in susceptible individuals. However, the prognostic value of fever-induced Brugada ECG pattern remains unclear. We analyzed the clinical and extended long-term follow-up data of consecutive febrile patients with a type 1 Brugada ECG presented to the emergency department. A total of 21 individuals (18 males; mean age, 43.7 ± 18.6 years at diagnosis) were divided into symptomatic (resuscitated cardiac arrest in one, syncope in two) and asymptomatic (18, 86%) groups. Sustained polymorphic ventricular tachycardias were inducible in two patients with previous syncope. All 18 asymptomatic patients had no spontaneous type 1 Brugada ECG recorded at second intercostal space and no family history of sudden death. Among asymptomatic individuals, 4 had a total 12 of repeated non-arrhythmogenic febrile episodes all with recurrent type 1 Brugada ECGs, and none had a ventricular arrhythmic event during 116 ± 19 months of follow-up. In the symptomatic group, two had defibrillator shocks for a new arrhythmic event at 31- and 49 months follow-up, respectively, and one without defibrillator therapy died suddenly at 8 months follow-up. A previous history of aborted sudden death or syncope was significantly associated with adverse outcomes in symptomatic compared with asymptomatic individuals (log-rank *p* < 0.0001). In conclusion, clinical presentation or history of syncope is the most important parameter in the risk stratification of febrile patients with type 1 Brugada ECG. Asymptomatic individuals with a negative family history of sudden death and without spontaneous type 1 Brugada ECG, have an exceptionally low future risk of arrhythmic events. Careful follow-up with timely and aggressive control of fever is an appropriate management option.

## 1. Introduction

Brugada syndrome is a distinct arrhythmogenic genetic disorder characterized by an ECG pattern of coved-type ST-segment elevation in the right precordial leads (V_1_–V_3_) at an increased propensity for ventricular fibrillation and the risk of sudden cardiac death [1]. Several non-genetic factors or conditions have been reported to induce Brugada ECG pattern or instead to unmask the concealed form of Brugada syndrome [2,3,4,5,6,7,8]. The most frequent finding associated with the Brugada ECG pattern is fever. A febrile state may provoke a Brugada ECG pattern and trigger ventricular tachyarrhythmias in susceptible individuals. However, the prognostic value of Brugada ECG pattern changes observed only during febrile illness remains unclear. Conflicting study results on this association make it uncertain whether fever-induced Brugada ECG pattern bears any relation to the risk of sudden cardiac death. Junttila et al. [4] reported that in most febrile cases with a typical Brugada ECG pattern, sudden cardiac death or malignant arrhythmias developed shortly after the onset of fever regardless of the existence of a predisposing genetic base. In contrast, Adler et al. [8] demonstrated that none of the eight patients presenting with a fever-induced Brugada ECG pattern had arrhythmic events during a 30 ± 13-month follow-up period without antiarrhythmic therapy. Additionally, risk stratification and management of asymptomatic febrile patients with Brugada ECG patterns remain controversial. Some investigators have suggested that patients presenting with a Brugada ECG pattern are at considerably higher risk of sudden cardiac death and that Brugada ECG pattern should be considered a medical emergency [4,7]. However, opponents of this view argue that fever may cause a transient Brugada ECG pattern in susceptible patients who do not have the genetically defined syndrome [2]. Previous studies have demonstrated that asymptomatic individuals, and in particular individuals with only transient ECG abnormalities, are at low risk of sudden cardiac death [9,10]. Therefore, the purpose of this study was to define the clinical relevance and evaluate the risk of ventricular arrhythmias associated with fever-induced Brugada ECG pattern in different clinical situations, and to present extended long-term follow-up data on clinical outcomes in the largest ever reported series of consecutive individuals with fever-induced Brugada ECG pattern.

## 2. Methods

### 2.1. Study Population

Consecutive patients admitted to a tertiary university hospital emergency department between May 2009 and May 2014 were screened by weekly review of ECG recordings and all consecutive febrile patients (defined as body temperature >38 °C by tympanic thermometer probe) with a type 1 Brugada ECG pattern characterized by a right bundle branch block and a high take-off >2 mm coved ST-segment elevation, followed by a negative T wave in at least 2 right precordial leads (V_1_–V_3_) according to the Second Brugada Consensus Conference criteria [3] were included as the analytic sample. Patients with only type 2 or 3 Brugada ECG of lesser degrees or different contours of ST-segment elevation (“saddleback” rather than “coved”) were excluded because these findings are not diagnostic and often interpreted inconsistently. All available ECGs of the included patients were independently analyzed by two electrophysiologists (CF Tsai and YT Chuang) and reevaluated after their fevers had subsided. Normalization of Brugada pattern on ECG after resolution of fever confirms the diagnosis of fever-induced Brugada ECG pattern (Figure 1). Patients with a persistent type 1 Brugada ECG after the fever subsided were excluded. In addition, patients with other conditions previously reported to cause similar ECG abnormalities, such as atypical right bundle branch block, acute pericarditis, dissecting aortic aneurysm, electrolyte abnormalities, or mechanical compression of the right ventricular outflow tract, were also excluded [3].

### 2.2. Study Design and Ethical Considerations

In this prospective cohort study, included patients were categorized into two groups: a symptomatic group that included patients with a history of cardiac arrest or syncope of suspected arrhythmia etiology, and an asymptomatic group that included patients without a personal history of sudden cardiac death or syncope. The human research committee of the study hospital approved the study protocol. All included patients provided signed informed consent to participate in the study.

### 2.3. Clinical Follow-Up

All patients diagnosed with fever-induced Brugada ECG pattern were prospectively evaluated and followed, at least once, in our institution’s arrhythmic clinic after they were discharged directly from the emergency department or from their hospitalization for the index febrile illness.

Demographic and clinical data of interest were (1) age, (2) gender, (3) presenting symptoms, (4) febrile illness diagnosis, (5) family history of sudden cardiac death, and (6) history of syncope or other arrhythmic symptoms (e.g., severe palpitation, etc.). Routine examinations, including at least echocardiography, excluded any underlying structural heart disease, and laboratory tests excluded acute ischemia and metabolic or electrolyte abnormalities. All patients underwent ECG recordings during the afebrile state at standard lead locations and the high lead position with right precordial leads (V_1–2_) placed at the second intercostal space to check for the presence of spontaneous type 1 Brugada ECG [3]. All patients were advised to undergo treadmill exercise testing and 24 h Holter monitoring for arrhythmia evaluation. Genetic testing for mutation analysis of the SCN5A gene was also suggested to all included patients. Drug challenge tests and invasive electrophysiological studies were recommended for individuals with positive family or syncopal history, or a spontaneous type 1 Brugada ECG, which may indicate the possible diagnosis of Brugada syndrome. Propafenone or flecainide was used for the class I antiarrhythmic drug challenge [3,11]. The programmed electrical stimulation protocol included a maximum of 3 ventricular extra-stimuli delivered from two ventricular sites (right ventricular apex and outflow tract), with the endpoint being the inducibility of sustained ventricular tachyarrhythmias causing syncope or requiring emergency intervention. An implantable cardioverter-defibrillator (ICD) was strongly recommended for individuals with a history of cardiac arrest and those with inducible ventricular arrhythmias.

Asymptomatic patients with a negative syncope workup or family history of sudden cardiac death were asked to seek a routine cardiac evaluation once annually; if unavailable, all patients were then contacted by phone or letter to determine the status of their arrhythmic symptoms and whether the patient was still alive. All patients were strongly recommended to avoid certain medications responsible for Brugada ECG pattern changes and to receive urgent treatment (e.g., oral paracetamol or a cold compress) for alleviating fever. During the follow-up period, patients were considered to have an arrhythmic event if sudden cardiac death occurred, or when appropriate, ICD shocks or sustained ventricular tachyarrhythmias were documented.

### 2.4. Statistical Analysis

Paired and unpaired data and survival curve data were analyzed using the SPSS software package (SPSS, Chicago, IL, USA). The time to the first arrhythmic event was depicted with the Kaplan–Meier estimate of the survival function. The difference between the survival curves was tested by the log-rank statistics. Continuous variables are presented as means ± SDs and compared using an unpaired *t* test or Mann–Whitney U test, depending on data distribution. A value of *p* < 0.05 was considered statistically significant.

## 3. Results

### 3.1. Clinical Characteristics of Patient Population

A total of 24 consecutive febrile patients with a type 1 Brugada ECG pattern presented to the emergency department during the study period. Three patients aged 73 ± 4 years were excluded, including one male with gastric cancer with liver metastasis, one male with multiple hepatocellular carcinomas, and one female with pulmonary tuberculosis who experienced in-hospital mortality due to hepatic failure in one and severe sepsis in two, respectively. The study patient population consisted of 21 patients (18 males) with a mean age of 43.7 ± 18.6 years at diagnosis (median, 42 years; range, 21 to 83 years). Mean temperature at the time of the type 1 Brugada ECG recording was 38.9 ± 0.8 °C (range, 38–41 °C) with a mean heart rate of 108 ± 17 beats/min (range, 80–157 beats/min) and a white blood count of 13,080 ± 4885 cells/mm^3^ (range, 5540–21,440 cells/mm^3^) at presentation.

Of the 21 patients with fever-induced Brugada ECG pattern, 19 (90%) presented with symptoms related to fever or an underlying febrile illness, including infectious diseases in 16 patients (3 pneumonia, 3 acute appendicitis, 2 upper respiratory infection, 2 biliary tract infection, 2 urinary tract infection, 1 liver abscess, 1 influenza, 1 infectious diarrhea, 1 infectious colitis), heatstroke in 1 patient, phenytoin-induced Stevens–Johnson disease in 1, and carbon monoxide intoxication in 1 patient (Figure 2). All fever-induced Brugada ECG patterns were incidental findings. Two patients were suspected of having acute coronary syndrome based on an ECG finding of right precordial leads ST-segment elevation; none underwent emergent coronary arteriography because of the absence of any cardiac symptoms and a normal cardiac enzymes level. However, none of the presented ECG cases was identified as type 1 Brugada ECG pattern at the emergency department. Most patients (15 of 19, 79%) were admitted to the medicine or surgery departments to treat their underlying febrile diseases. One patient (male aged 29 years) with the diagnosis of phenytoin-induced Stevens–Johnson disease reported one syncope event leading to a traffic accident and head injury one month prior to admission. He also had a spontaneous type 1 Brugada ECG pattern while afebrile previously. A propafenone drug challenge failed to induce any significant ECG changes. Sustained polymorphic ventricular tachycardia was induced from the right ventricular apex with up to triple extra-stimuli, and an ICD was implanted subsequently under the impression of Brugada syndrome during his index hospitalization. The remaining 18 patients underwent outpatient follow-up in the arrhythmic clinic, and no previous arrhythmias or episodes of syncope or sudden cardiac death were noted in their clinical and familial history. No patients showed up with type 1 Brugada ECG pattern during the afebrile state, and only two patients demonstrated a type 3 Brugada ECG pattern when ECGs were recorded at the high (second) right intercostal space. The results of echocardiography in all and ambulatory Holter monitoring in six patients were unremarkable. All 18 patients refused to undergo an antiarrhythmic drug challenge test or invasive electrophysiologic study, or genetic testing to look for known mutations implicated in Brugada syndrome. 

One patient (male aged 79 years) presented with syncope and another patient (male aged 57 years) experienced an episode of ventricular fibrillation and aborted cardiac arrest at presentation to the emergency department. Given their acute cardiac symptoms with new ST-segment elevation, the initial evaluation of these patients was focused on acute pulmonary embolism, aortic dissection, and acute coronary syndrome (Figure 3). Both patients had a normal chest computed tomography study and emergent cardiac catheterization, which revealed normal coronaries and left ventricular systolic function. No family or personal history of syncope or sudden death was found in either patient. In the 79-year-old syncopal patient, the Brugada ECG pattern resolved with defervescence and re-emerged during the propafenone provocation test, and the electrophysiologic testing with standard programmed ventricular stimulation techniques was positive for arrhythmia induction. Although this patient was offered defibrillator implantation, he opted for conservative management. The other patient fulfilled the diagnostic criteria of Brugada syndrome and subsequently received ICD placement during his index hospitalization.

The characteristics of patients between the symptomatic group (with a history of syncope or sudden cardiac death, *n =* 3) and the asymptomatic group (*n =* 18) are summarized in Table 1. The age at presentation, body temperature, laboratory data (including WBC, CRP, Na/K) were not significantly different between the symptomatic and asymptomatic groups (*p* > 0.05).

### 3.2. Effect of Fever on ECG Parameters

ECGs during fever and in the afebrile state were available for analysis in all 21 patients. ECG parameters were measured and compared between the symptomatic group (with a history of syncope or sudden cardiac death, *n =* 3) and the asymptomatic group (*n =* 18) (Table 2). During fever, the heart rate was significantly higher than that at afebrile status in the asymptomatic group (108 ± 20 beats/min vs. 77 ± 7 beats/min; *p* < 0.001) and in the symptomatic group (105 ± 1 beats/min vs. 74 ± 4 beats/min; *p* < 0.001), while the PR interval was significantly shorter than that at afebrile status in the asymptomatic group (159 ± 27 ms vs. 180 ± 21 ms; *p* < 0.05) and in the symptomatic group (149 ± 6 ms vs. 165 ± 13 ms; *p* < 0.05). However, the increase in heart rate and the decrease in PR interval during fever were not significantly different between the symptomatic and asymptomatic groups (*p* > 0.05). QRS duration during fever was not significantly different from that at afebrile status in both the asymptomatic group (102 ± 11 ms vs. 97 ± 8 ms; *p* = 0.09) and the symptomatic group (107 ± 14 ms vs. 90 ± 8 ms; *p* = 0.14). The corrected QT interval during fever was not significantly different from that at afebrile status in both the asymptomatic group (431 ± 42 ms vs. 436 ± 40 ms; *p* = 0.74) and the symptomatic group (453 ± 29 ms vs. 426 ± 24 ms; *p* = 0.29). Changes in QRS duration and corrected QT intervals during fever were not significantly different between the symptomatic and asymptomatic groups (*p* > 0.05).

### 3.3. Follow-Up Data

#### Recurrence and Reversibility of Fever-Induced Brugada ECG Pattern

Four patients had a recurrence of type 1 Brugada ECG morphology with repeated febrile episodes (recurrence once in 1, twice in 2, three times in 1). All Brugada ECG pattern anomalies were evident as long as fever was present and vanished once the temperature returned to normal (Figure 4). Neither the etiology nor the height of the fever was able to predict the recurrence and the configuration of the Brugada pattern in these recurrent febrile episodes. The Brugada ECG pattern was not arrhythmogenic in any episode. These patients opted not to pursue any further testing but to return for follow-up of their abnormal ECG findings on an outpatient basis. Given the absence of cardiac symptoms at the recurrent episodes and resolution of the Brugada ECG pattern after the fever subsided, patients were discharged and advised to remember that timely and aggressive control of fever is imperative.

### 3.4. Long-Term Outcomes

The mean follow-up period for the entire study population was 116 *±* 19 months (range, 84–144 months). During the follow-up period, only one patient with prior cardiac arrest experienced one-time ICD shock for ventricular fibrillation in the afebrile state at a 49-month follow-up. In two patients identified after syncope, one had documented recurrent ventricular tachyarrhythmias requiring ICD interventions (3 events at a 105-month follow-up) and the patient without ICD therapy died unexpectedly at an 8-month follow-up. Among all 18 asymptomatic individuals followed without antiarrhythmic therapy, none had a ventricular arrhythmic event. Differences in outcomes (free of sudden death or ventricular fibrillation events) between the two groups are shown in Figure 5. In the presence of low statistical power because of the limited number of events, having a previous history of aborted sudden cardiac death or syncope was significantly associated with adverse outcomes in symptomatic compared with asymptomatic individuals (log-rank *p* < 0.0001).

## 4. Discussion

This single-center prospective study demonstrated the outcomes of a cohort of 21 consecutive patients with fever-induced Brugada ECG pattern and, which, to the best of our knowledge, features the longest follow-up to date (median 10-year follow-up; range, 84–144 months). Prospective data were evaluated, reporting the “true” outcomes of fever-induced Brugada ECG pattern in febrile subjects. Results of the present study suggest that the incidental finding of type 1 Brugada ECG in otherwise healthy individuals is seldom recognized by emergency or general physicians. Among subjects with cardiac symptoms, prominent ST elevation at right precordial leads is easily misdiagnosed as an acute coronary syndrome. Our results suggest that clinical presentation or history of aborted cardiac death or syncope is the most important parameter in the risk stratification of febrile patients with type 1 Brugada ECG. In the present study, asymptomatic patients without a family history of sudden death appeared to have a good prognosis. Additionally, the recurrence and reversibility of type 1 Brugada ECG pattern associated with repeated febrile episodes confirm the critical role of fever in uncovering this ECG phenomenon in susceptible individuals. However, the non-arrhythmogenic nature of the ECG phenotype was maintained over time in asymptomatic patients without a history of syncope or sudden cardiac death.

### 4.1. Recognition of Type 1 Brugada ECG Pattern in Febrile Subjects

Brugada syndrome is a distinct arrhythmogenic disorder widely recognized as an important cause of sudden death in the young and is diagnosed when a patient has a characteristic Brugada type ECG consisting of a coved-type ST elevation in the leads V_1-3_ and documented ventricular tachyarrhythmias or history consistent with ventricular tachyarrhythmias, such as syncope or sudden cardiac death. Therefore, diagnosis of Brugada syndrome is based mainly on the recognition of a putative Brugada pattern on an individual’s ECG. The Brugada pattern, on the other hand, can be unmasked by fever and resolve after fever subsides [2,3,4,7,8]. However, over the past few decades, in the emergency and general internal medicine literature, studies on recognition of fever-induced Brugada ECG patterns have been very limited [12,13,14,15,16,17,18]. We confirmed that this peculiar ECG pattern is left unrecognized in patients without cardiovascular symptoms by emergency and internal medicine physicians and surgeons. Instead, the prominent ST elevation in the Brugada ECG pattern may mimic ST-segment elevation myocardial infarction, posting a clinical challenge to emergency and general physicians. This report highlights the importance of recognizing the characteristic Brugada ECG pattern and considering it to be a diagnostic clue for Brugada syndrome. Fever is a common clinical problem and may act as a precipitant for increased susceptibility to potentially fatal ventricular arrhythmias in Brugada syndrome [3,4,7,12]. Therefore, if the Brugada ECG pattern is identified in febrile patients, a detailed patient and family history of syncope or sudden cardiac death is deemed necessary for risk assessment. In the present study, fever-induced Brugada ECG pattern in individuals without clinical or ECG indications of the genetic Brugada abnormality appears to be a relatively benign condition. Among 18 asymptomatic patients, none had a spontaneous type 1 Brugada ECG in the baseline afebrile state and all of them were free of any arrhythmic events at a median follow-up of 116 months. If someone presents with arrhythmic events or syncope during fever, the Brugada syndrome should be considered as a possible diagnosis. Physicians should diagnose syncopal patients with fever cautiously, and repetitive ECG recordings at the higher intercostal spaces may be helpful for enabling a diagnosis [19]. Recording ECGs at higher right intercostal spaces increases the sensitivity of Brugada ECG pattern detection [3,20]. Similar to a prior study [21], only ~10% of subjects in the present study demonstrated a non-diagnostic type 3 Brugada ECG pattern when ECGs were recorded at high right intercostal spaces in the baseline afebrile state.

### 4.2. Comparison with Previous Studies

The independent clinical significance of the fever-induced Brugada ECG pattern remains unknown. In asymptomatic febrile subjects with a Brugada ECG pattern, the risk of serious arrhythmic events is not well defined. Consistent with the results of an Italian community-based study [22], the risk of future arrhythmic events in asymptomatic febrile individuals without a history of syncope or cardiac arrest is very low, most likely similar to that in the general population. Programmed electrical stimulation seems valuable in patients with a previous syncope. In the present study, two febrile patients with a Brugada ECG pattern and a personal history of syncope were found to have inducible ventricular fibrillation on programmed ventricular stimulation and also experienced a first arrhythmic event at 8- and 31-month follow-up. These results reinforce the notion that clinical presentation is the most important parameter in the risk stratification of febrile patients with a Brugada ECG pattern.

Several studies have demonstrated that fever unmasks or promotes the characteristic Brugada ECG pattern and precipitates ventricular arrhythmias. Junttila et al. [4] reported that in most febrile cases in their series with a typical Brugada ECG pattern, sudden cardiac death or malignant arrhythmias developed shortly after the onset of fever regardless of the existence of a predisposing genetic base. In that study, among the 16 patients with fever-induced type 1 Brugada ECGs, 10 had a history of cardiac arrest or syncope. Additionally, Amin et al. [7] demonstrated that fever precipitated malignant arrhythmias in 18% of patients presenting with cardiac arrest in symptomatic Brugada syndrome. However, in both of these studies, the cardiac arrest occurred mostly at the time of fever presentation, not during the follow-up period. This selection bias may overestimate the long-term arrhythmic risks for febrile patients with an incidental finding of the Brugada ECG pattern. In a large cohort of patients with documented Brugada syndrome, Michowitz et al. [23] showed that ~6% of arrhythmic events were associated with fever and, among these fever-related events, 83% occurred in Caucasian males and 80% presented with aborted cardiac death. A syncopal history and spontaneous type 1 Brugada ECG were noted in 40% and 71%, respectively, of patients with fever-related arrhythmic events. That international multicenter study also confirmed that the involvement of Asians in fever-related arrhythmic events was extremely rare, especially in children. The previous studies described above concluded that patients with Brugada syndrome who develop a fever-induced type 1 ECG are at risk of arrhythmic events. However, in the present study, asymptomatic patients without syncope history or spontaneous type 1 Brugada ECG or family history of sudden cardiac death seldom conform to the diagnosis of Brugada syndrome. The fever-induced Brugada ECG pattern in asymptomatic individuals may be regarded as merely an ECG variant.

A prevalence of 2–4% of fever-induced Brugada ECG patterns in febrile patients referred to the emergency department was reported in two studies, although it was not assessed in this study. Adler et al. [8] reported 8 consecutive patients with fever-induced Brugada ECG patterns who were asymptomatic and remained free of arrhythmic events during 30 months of follow-up. Another study from an endemic area of Brugada syndrome showed that the prevalence of fever-induced Brugada ECG pattern was even higher, up to 5.3%, in febrile male subjects [19]. This estimate, which is dozens of times higher than the known prevalence of Brugada syndrome in the general population, highlights the importance of features distinguishing fever-induced Brugada ECG changes and Brugada syndrome [3]. Mizusawa et al. [24] found that of the 88 asymptomatic patients with fever-induced type 1 Brugada ECG at baseline, 2 patients without evidence of spontaneous type 1 ECG, family cardiac arrest history, or SCN5A mutation experienced sudden cardiac death at the 10- and 75-month follow-up, respectively. Enrollment of non-consecutive cases from an international registry may account for the discrepancies between the study results. The higher event rates in the international registry may be attributable to selection bias stemming from the inclusion of more severely affected patients and families from the 1990s when the syndrome was first described. This study enrolled the largest cohort to date of consecutive febrile patients with Brugada ECG patterns in a single center and reported a very low incidence of arrhythmic events in asymptomatic patients followed for an average of 10 years.

Amin et al. [7] found that, regardless of the cause, fever markedly increased the mean PR/QRS intervals, QTc duration, and ST-segment amplitude in leads V_1_ and V_2_ for patients with Brugada syndrome. However, inconsistent with a recent study on the relevant ECG markers associated with fever, we found that the PR interval was significantly shorter in the febrile than the afebrile state in both the asymptomatic and symptomatic groups, accompanied by a significant increase in heart rate [24]. Only one asymptomatic patient with diagnosed ruptured acute appendicitis had PR interval prolongation compared with that in the afebrile state (240 ms vs. 190 ms). Febrile illness is a stress-based condition clinically, occurring at an increase in sympathetic activity leading to an increase in heart rate and atrioventricular nodal conduction. Additionally, autonomic influences seem to play an important role in the modulation of the electrophysiology and arrhythmogenesis of Brugada syndrome [25]. ST-segment elevation in the Brugada ECG pattern is mitigated by the administration of beta-adrenergic agonists and is enhanced by parasympathetic agonists such as acetylcholine in experimental and clinical investigations [25,26]. Taken together, these observations show that fever triggers the Brugada ECG phenomenon through the modulatory effect of temperature itself rather than autonomic input.

### 4.3. Recurrence and Reversibility of Fever-Induced Brugada ECG Pattern

In this study, 4 of 21 patients who presented to the emergency department had recurrent febrile episodes with a total of 12 available ECGs, all of which showed type 1 Brugada ECG pattern regardless of the level and etiology of pyrexia. Furthermore, their classic findings of fever-induced Brugada ECG patterns were reversible with clinical defervescence. These ECG temporal changes demonstrate the direct modulating effect of temperature on the mechanism of the Brugada ECG phenomenon in susceptible individuals. In the setting of Brugada syndrome, mutant sodium channels are shown to be temperature dependent with further impairment occurring at elevated temperature, leading to more evident ECG abnormalities and predisposing to arrhythmias [27]. Patients demonstrating fever-induced Brugada ECG pattern are also likely to be genetically predisposed, although it is not clear whether or to what extent a genetic predisposition may be involved. Insight from cellular electrophysiology suggests that accentuation of the right ventricular action potential notch by a rebalancing of active currents at the end of phase 1 may give rise to the typical Brugada ECG without creating an arrhythmogenic substrate [28,29]. It is possible that a febrile state may modulate the functional expression of ionic currents responsible for the dynamic ECG changes. Fever-induced Brugada ECG pattern may be due to increased susceptibility to fever-induced ECG abnormalities, possibly as a result of an increase in a latent ion channel dysfunction similar to that in drug-induced long QT syndrome. Fever simply unmasks the type 1 Brugada pattern in carriers of this genetically determined arrhythmic disorder. However, further evidence is needed to confirm this postulation. Additionally, recent genetic studies provide new insights on the existence of a pathogenetic link between Brugada syndrome and arrhythmogenic cardiomyopathy. Brugada syndrome may be associated or overlapping with structurally yet not phenotypically expressed cardiomyopathies, such as right ventricular arrhythmogenic cardiomyopathy, hypertrophic cardiomyopathy, and Lamin A/C cardiomyopathy [30,31]. Scheirlynck et al. [32] reported that patients with overlapping phenotypes were associated with a trend toward higher arrhythmic risk. None of the asymptomatic patients in our study underwent cardiac magnetic resonance image or genetic study to address this issue at the follow-up period, and they really showed an exceptionally low future risk of arrhythmic events.

Despite the non-arrhythmogenic nature of all recurrent fever-induced Brugada ECG patterns in asymptomatic patients over a decade-long follow-up period, the lifetime probability of a cardiac event in these patients is not well defined. Careful follow-up is strongly recommended with prompt and aggressive treatment of fever with antipyretics and cold compresses. In addition to being exposed to fever, the Brugada ECG pattern may be unmasked by numerous medications, alcohol and cocaine intoxication, hypokalemia, or other physiologic disruptions [2,3,4,5]. The coexistence of multiple trigger factors may possibly cause a more pronounced Brugada ECG phenotype and increase the risk of ventricular arrhythmias. Physicians should be aware of these possible triggers and educate affected patients to avoid them. These cautions were communicated to all of our patients in follow-up at arrhythmic clinics. ICD implantation is the proven effective treatment modality for aborted cardiac arrest survivors and patients with a history of syncope and documented ventricular arrhythmia [3]. However, in asymptomatic individuals with Brugada syndrome, ICD implantation is known to have high complications associated with the procedure and is therefore not recommended, as it has no mortality benefit [33].

### 4.4. Study Limitations

This study has several limitations, including that febrile groups included in the present study may not necessarily represent the general population because all of them were initially evaluated at an emergency department. ECG is usually not recorded in an otherwise healthy non-cardiac patient presenting to the outpatient clinic with a fever. Additionally, only adults (the youngest man, 21 years old at presentation) were studied because of the independent pediatric emergency room in our hospital. Fever is the most frequent trigger for syncope and sudden death among children with occult Brugada syndrome [34]. However, in a multicenter cohort of Brugada syndrome, Asians with fever-related arrhythmic events were much older than their Caucasian counterparts, and the youngest was 25 years old [23]. Moreover, we only included subjects admitted to the hospital; however, it may be that death during the febrile state in patients not admitted to the hospital had some share of Brugada pattern subjects. This would conflict with the risk of acute cardiac events associated with a febrile Brugada pattern but very unlikely would affect the outcome of those who do not have acute events. One major limitation in evaluating the asymptomatic patient group is that none of these patients chose to undergo the recommended genetic testing, drug challenge testing, or electrophysiological study, which may have further elucidated their condition. The role of programmed ventricular stimulation in risk stratification also has been controversial. In fact, some authors have reported that the inducibility of ventricular tachyarrhythmias by programmed electrical stimulation was not a significant predictor of future arrhythmic events [35,36]. Finally, the results of this study are limited by the low statistical power of the small case series and the limited number of events. Nevertheless, findings of a true consecutive cohort certainly add clinically relevant insights in managing patients with fever-induced Brugada ECG patterns.

## 5. Conclusions

The present study reported data of a cohort of consecutive patients presenting with fever-induced type 1 Brugada ECG pattern with the longest follow-up reported to date. The lack of cardiac symptoms (syncope, sudden cardiac death) at presentation among asymptomatic patients, with negative personal and family history and without spontaneous type 1 Brugada ECG, suggests an exceptionally low future risk of arrhythmic events. Careful follow-up with a recommendation of timely and aggressive control of the fever is an appropriate option for this patient population.

## Figures and Tables

**Figure 1 jcm-10-04997-f001:**
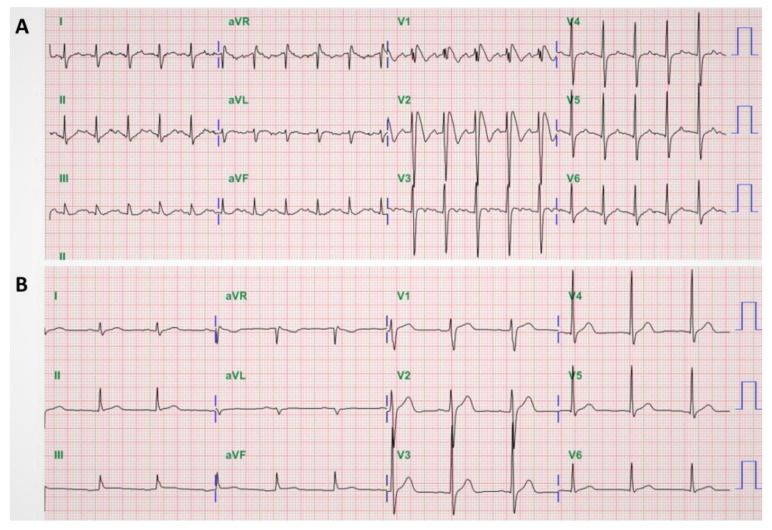
Fever-induced Brugada electrocardiography (ECG) pattern. A 45-year-old man was admitted with diarrhea and abdominal cramps for 2 days. He was febrile (temperature, 40 °C) with tachycardia (pulse, 127 beats/min): (**A**) ECG on admission revealed a right bundle-branch block with coved ST-segment elevation in V_1_–V_2_ followed by a negative T wave; (**B**) repeated ECG after fever resolved showed normalization of the ST-segment elevation, as well as disappearance of the right bundle-branch block.

**Figure 2 jcm-10-04997-f002:**
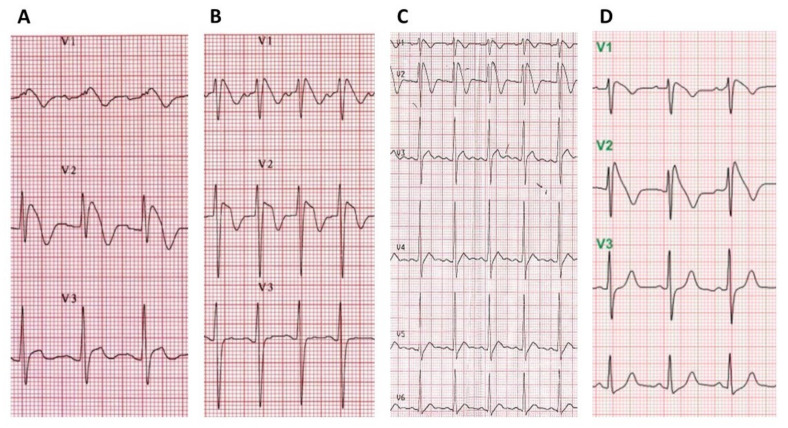
Fever-induced Brugada ECG pattern in different clinical situations. Typical Brugada ECG pattern during the febrile state is evident in different conditions: (**A**) acute tonsilitis in a 25-year-old man (temperature, 39 °C); (**B**) heatstroke in a 24-year-old man (temperature, 41 °C); (**C**) phenytoin-induced Stevens–Johnson disease in a 29-year-old man (temperature, 39.4 °C); (**D**) carbon monoxide intoxication in a 25-year-old female patient (temperature, 38 °C).

**Figure 3 jcm-10-04997-f003:**
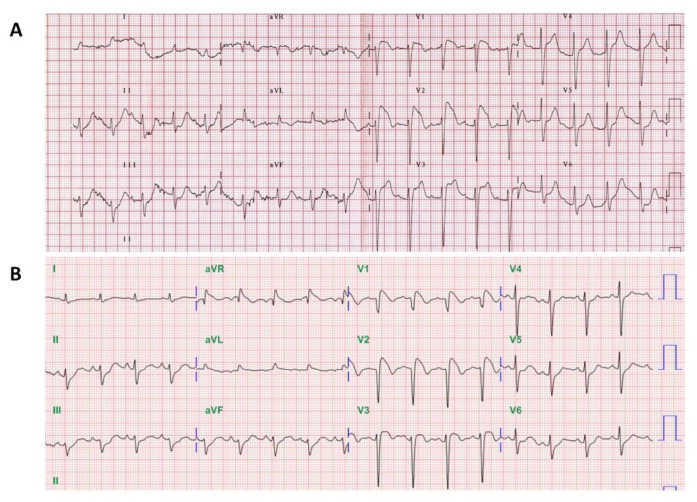
Fever-induced Brugada ECG pattern masquerading as acute coronary syndrome. ECG demonstrated prominent ST-segment elevations in leads V1-3 mimicking anterior wall ST-elevation myocardial infarction in (**A**) a 79-year-old man admitted due to syncope on the roadside and (**B**) a 57-year-old man with out-of-hospital cardiac arrest after resuscitation and recovery of spontaneous circulation.

**Figure 4 jcm-10-04997-f004:**
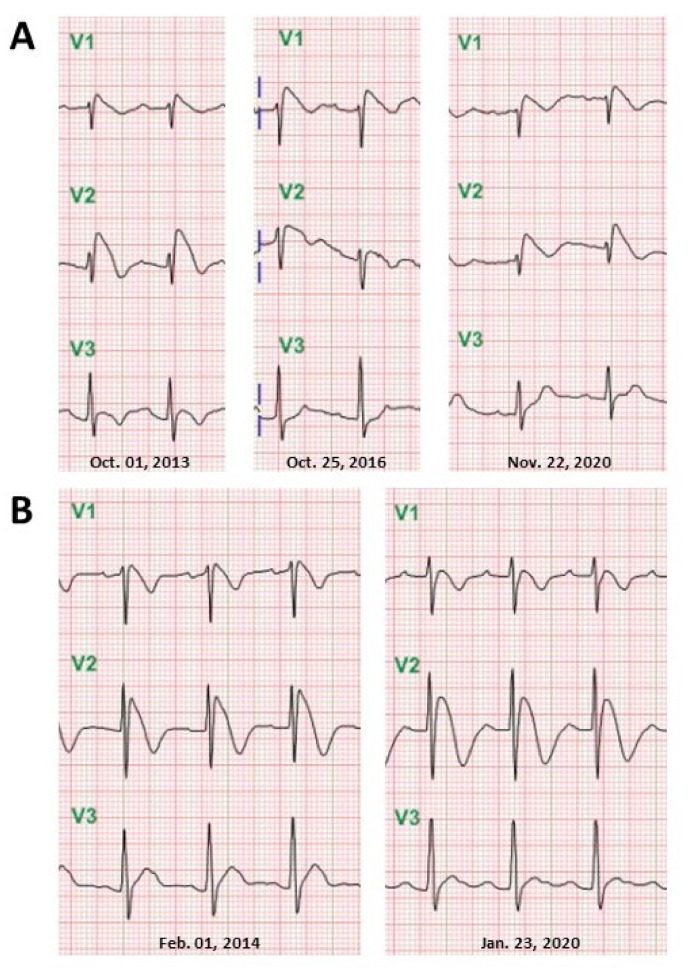
Recurrence of fever-induced Brugada ECG pattern. Admission ECGs displaying recurrences of coved-type ST-segment elevation accompanying repeated febrile episodes in (**A**) an 83-year-old man on 1 October 2013: *Klebsiella pneumoniae* sepsis (temperature, 39.8 °C); on 25 October 2016: upper respiratory tract infection (temperature, 39.2 °C); on 22 November 2020: pneumonia (temperature, 38.5 °C); (**B**) a 21-year-old man on 1 February 2014: infectious diarrhea (temperature, 38.3 °C) and on 23 January 2020: infectious diarrhea (temperature, 38.7 °C).

**Figure 5 jcm-10-04997-f005:**
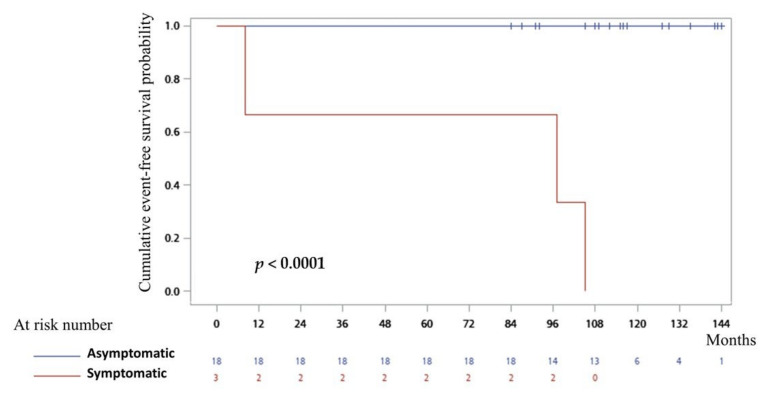
Cumulative survival curves during the follow-up period. Kaplan–Meier analysis of arrhythmic events (sudden cardiac death or documented ventricular fibrillation) during the follow-up depending on clinical presentation with symptomatic (aborted sudden cardiac death or syncope of suspected arrhythmic origin) or asymptomatic individuals when a fever-induced Brugada ECG pattern was identified. The difference between the two groups was statistically significant (log-rank *p* < 0.0001).

**Table 1 jcm-10-04997-t001:** Characteristics of patients between the two groups.

Characteristics	Asymptomatic (*n* = 18)	Symptomatic (*n* = 3)	*p*
Gender	M (15)/F (3)	M (3)	
Age at diagnosis (yr)	42 ± 18	55 ± 25	0.26
Temperature (°C)	38.8 ± 0.8	39.1 ± 0.6	0.54
WBC (cells/mm^3^)	12,908 ± 5046	14,113 ± 4503	0.70
C-reactive protein (mg/dL)	8.9 ± 8.4	13.5 ± 6.6	0.38
Sodium (mmol/L)	136 ± 3	137 ± 2	0.80
Potassium (mmol/L)	3.8 ± 0.4	3.7 ± 0.8	0.19
Cardiac symptoms	None	Cardiac arrest (1) syncope (2)	
Programmed ventricular stimulation	NA (0/18)	Polymorphic VT (2/2)	
ICD therapy	None	Two	

Values are presented as mean ± SD. ICD = implantable cardioverter-defibrillator; NA = not available; VT = ventricular tachycardia; WBC = white blood count.

**Table 2 jcm-10-04997-t002:** ECG parameters during fever and in the afebrile state between the two groups.

Variable	Asymptomatic (*n* = 18)	Symptomatic (*n* = 3)	*p*
HR febrile	108 ± 20	105 ± 1	0.77
HR afebrile	77 ± 7	74 ± 4	0.41
Δ HR	31 ± 17	31 ± 3	0.99
PR febrile	159 ± 27	149 ± 6	0.52
PR afebrile	180 ± 21	165 ± 13	0.27
Δ PR	−20 ± 23	−18 ± 9	0.87
QRSd febrile	102 ± 11	107 ± 14	0.54
QRSd afebrile	97 ± 8	90 ± 8	0.21
Δ QRSd	5.8 ± 13	17 ± 14	0.18
QTc febrile	431 ± 42	453 ± 29	0.40
QTc afebrile	436 ± 40	426 ± 24	0.70
Δ QTc	−4.6 ± 44.86	26.3 ± 12.7	0.26

Values are presented as mean ± SD. HR = heart rate (beats/min); PR = PR interval (msec); QRSd = QRS duration or width (msec); QTc = corrected QT interval (msec); Δ = the difference of parameters between febrile and afebrile state.

## Data Availability

The data presented in this study are available on request from the corresponding author.

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
