# Peer review of "Long-Term Prognosis of Febrile Individuals with Right Precordial Coved-Type ST-Segment Elevation Brugada Pattern: A 10-Year Prospective Follow-Up Study"

_jcm, 2021, doi:10.3390/jcm10214997_

Round 1
Reviewer 1 Report
This article addresses a very important aspect in the management of patients admitted to the hospital with a type 1 Brugada pattern during fever.
They report a fairly good outcome at long term follow-up for subjects without of history of syncope or aborted cardiac arrest, che remain the only predictive prognostic factor beyond all the clinical and electrocardiographic features in both the febrile and afebrile status.
The observation is numerically powered and the follow-up time is consistent to support their conclusions.
However, there are some methodological limitations that cannot be overcome:
- we have no figure of the prevalence of fever induced Brugada pattern, because only subjects admitted to the hospital are detected. It may be that death during febrile state in patients not admitted to the hospital has some share of Brugada pattern subjects. This would conflict with the risk of ACUTE cardiac events associated with a febrile Brugada pattern, but very unlikely would affect the outcome of those who do not have acute events. Please discuss
- no patient underwent MR scanning and genetic testing, This is mostly important because a Brugada pattern may be associated or overlapping with structurally yet not phenotypically expressed cardiomyopathy, such as Right ventricular Arrhtymogenic Cardiomyopathy, Hypertrophic Cardiomyopathy, Lamin A/C cardiomyopathy (see references in literature; Huang Cerrone, Nademanee, Farnè, Armaroli ): this aspect was not re-assessed at follow-up, so it should be discussed.
- The Europace Paper by Delise and Probst should be referenced as well at ref [9]
Reviewer 2 Report
This is a quite unique series of patients
Author Response
Responses to the reviewer 2
Thank you very much for your comments, which are very helpful to this manuscript, and we appreciate your valuable comments and for allowing us to revise this manuscript. Our study reports data of the largest ever reported series of consecutive patients presenting with fever-induced type 1 Brugada pattern ECG with the longest follow-up reported to date. This is a quite unique series of patients in clinical practice. The findings of a true consecutive cohort certainly add clinically relevant insights in managing patients with fever-induced Brugada pattern ECG. We believe that the readers are interested in our findings and the study results will significantly improve our knowledge of management of this patient population.
The above descriptions are the responses to your comments and suggestions.
Sincerely yours,
Chin-Feng Tsai, MD, PhD
Associate Professor, School of Medicine
Chung Shan Medical University
Department of Internal Medicine
Chung Shan Medical University Hospital
Taichung 40201, Taiwan
